# Exponential tilting of subweibull distributions

**F. William Townes**                                                *ftownes@andrew.cmu.edu*
*Department of Statistics and Data Science*
*Carnegie Mellon University*

**Reviewed on OpenReview:** *https://openreview.net/forum?id=BQBk11IE7I*

## Abstract

The class of subweibull distributions has recently been shown to generalize the important properties of subexponential and subgaussian random variables. We describe alternative characterizations of subweibull distributions, illustrate their application to concentration inequalities, and detail the conditions under which their tail behavior is preserved after exponential tilting.

## 1 Introduction

Subexponential and subgaussian distributions are of fundamental importance in the application of high dimensional probability to machine learning (Vershynin, 2018; Wainwright, 2019). Recently it has been shown that the subweibull class unifies the subexponential and subgaussian families, while also incorporating distributions with heavier tails (Vladimirova et al., 2020; Kuchibhotla & Chakrabortty, 2022). Informally, a q-subweibull ($q > 0$) random variable has a survival function that decays at least as fast as $\exp(-\lambda x^q)$ for some $\lambda > 0$. For example, the exponential distribution is 1-subweibull and the Gaussian distribution is 2-subweibull. Here, we provide alternative characterizations of the subweibull class and introduce a distinction between strictly and broadly subweibull distributions. As an example, the Poisson distribution is shown to be strictly subexponential ($q = 1$) but not subweibull for any $q > 1$. We demonstrate how subweibull properties can be used to prove Bernstein concentration inequalities in both heavy and light-tailed settings. Finally, we detail the conditions under which the subweibull property is preserved after exponential tilting.

To motivate this last result, consider the setting of adapting a model to a target distribution that differs from its training distribution. Maity et al. (2023) proposed an importance weighting strategy based on the assumption that the target distribution is well approximated by an exponential tilt of the training distribution. Our results will clarify particular settings where this approximation breaks down, and when one may transfer tail bounds from the training distribution to the target distribution.

## 2 Preliminaries

### 2.1 Laplace-Stieltjes transforms

**Definition 2.1.** The bilateral Laplace-Stieltjes transform (BLT) of a random variable $X$ with distribution function $F$ is

$$\mathcal{L}_X(t) = \mathbb{E}[\exp(-tX)] = \int_{-\infty}^{\infty} \exp(-tx) dF(x)$$

We do not restrict $X$ to be nonnegative or to have a density function. In the special case that $\mathcal{L}_X(t) < \infty$ for all $t$ in an open interval around $t = 0$, then $X$ has a moment generating function (MGF) which is $M_X(t) = \mathbb{E}[\exp(tX)] = \mathcal{L}_X(-t)$. The BLT can characterize the distribution even if the MGF does not exist.

**Lemma 2.0.1.** If the BLTs of random variables $X$ and $Y$ satisfy $\mathcal{L}_X(t) = \mathcal{L}_Y(t)$ for all $t$ in any nonempty open interval $(a, b) \subset \mathbb{R}$, not necessarily containing zero, then $X \stackrel{d}{=} Y$.

For a proof refer to Mukherjea et al. (2006). A random variable $X$ is considered subexponential iff the MGF exists (Vershynin, 2018). If $\mathcal{L}_X(t) = \infty$ for all $t > 0$ (respectively $t < 0$), $X$ is said to have a heavy left (respectively, right) tail (Nair et al., 2022). If a tail is not heavy it is said to be light. It is well known that the one-sided Laplace-Stieltjes transform characterizes nonnegative distributions (Feller, 1971). Lemma 2.0.1 shows that the BLT characterizes any distribution with at least one light tail.

## 2.2 Orlicz norms

An Orlicz function $\psi : \mathbb{R}_+ \mapsto \mathbb{R}_+$ is a nondecreasing, convex function with $\psi(0) = 0$. Unless explicitly stated below, we further restrict it to be strictly increasing.

**Definition 2.2.** The $\psi$-Orlicz norm of a random variable $X$ is given by

$$\|X\|_\psi = \inf\left\{t > 0 : \mathbb{E}\left[\psi\left(\frac{|X|}{t}\right)\right] \leq 1\right\}$$

where $\inf\{\emptyset\} = \infty$.

Since $\psi(|X|/t)$ is a decreasing function of $t$, it is clear that $\mathbb{E}[\psi(|X|/t)] \leq 1$ for all $t > \|X\|_\psi$. Like other norms, Orlicz norms have the following properties.

- Homogeneity: $\|aX\|_\psi = |a|\|X\|_\psi$

- Subadditivity: $\|X + Y\|_\psi \leq \|X\|_\psi + \|Y\|_\psi$

- Positive definiteness: $\|X\|_\psi = 0$ implies $X = 0$ almost surely.

Suppose $b \in \mathbb{R}$ is a constant. If $X$ is a degenerate random variable with $\Pr(X = b) = 1$, homogeneity implies $\|b\|_\psi = |b|/\psi^{-1}(1)$. Finiteness of the $\psi$-Orlicz norm is preserved under location-scale transformations. If $\|X\|_\psi < \infty$ and $a, b \in \mathbb{R}$, then

$$\|aX + b\|_\psi \leq |a|\|X\|_\psi + \|b\|_\psi < \infty$$

If $\|X\|_\psi = \infty$, then $\|aX\|_\psi = \infty$ as well. Here are some examples of Orlicz norms. Let $\psi(x) = x^p$ for $p \geq 1$. Then $\|X\|_\psi = \mathbb{E}[|X|^p]^{1/p}$ and $X \in L^p$ has finite moments up to order $p$ iff $\|X\|_\psi < \infty$. Here, we will focus primarily on the norm derived from the Orlicz function $\psi_q(x) = e^{x^q} - 1$, which is convex for $q \geq 1$.

**Definition 2.3.** The $\psi_q$-Orlicz norm ($q \geq 1$) of random variable $X$ is given by

$$\|X\|_{\psi_q} = \inf\left\{t > 0 : \mathbb{E}\left[\exp\left\{\left(\frac{|X|}{t}\right)^q\right\}\right] \leq 2\right\}$$

The condition that $\|X\|_{\psi_q} < \infty$ clearly implies that $|X|^q$ is subexponential. The following lemma, modified from Pollard (2024) is useful in establishing when a $\psi$-Orlicz norm is finite.

**Lemma 2.0.2.** Let $c_0, K_0 > 0$ be constants and $\psi$ an Orlicz function. The following are equivalent.

(a)
$$\mathbb{E}\left[\psi\left(\frac{|X|}{c_0}\right)\right] \leq K_0$$

(b)
$$\|X\|_\psi \leq c_0 \max\{K_0, 1\}$$

## 3 Subweibull random variables

**Definition 3.1.** A random variable $X$ is *q-subweibull* if $\mathbb{E}[\exp(\lambda^q |X|^q)] < \infty$ for some $\lambda > 0$. $X$ is *strictly q-subweibull* if the condition is satisfied for all $\lambda > 0$. If $X$ is q-subweibull but not strictly so, we refer to it as *broadly q-subweibull*.

The first part of this definition was also proposed by Kuchibhotla & Chakrabortty (2022) and by Vladimirova et al. (2020) using a parameterization equivalent to $1/q$. Clearly $X$ is (strictly) q-subweibull if and only if $|X|^q$ is (strictly) subexponential. As an example, the Laplace distribution is broadly 1-subweibull (ie broadly subexponential).

**Definition 3.2.** The *radius of convergence* of a q-subweibull random variable $X$ is defined by

$$R_q = \sup \{\lambda > 0 : \mathbb{E}[\exp(\lambda^q |X|^q)] < \infty\}$$

and if no such $\lambda > 0$ exists we adopt the convention that $R_q = 0$.

In the case of strictly q-subweibull distributions, $R_q = \infty$. $X$ has "heavy tails" (in the sense of Nair et al., 2022) iff it is not subexponential ($R_1 = 0$).

**Lemma 3.0.1.** Random variable $X$ with $\Pr(X < 0) \notin \{0, 1\}$ is q-subweibull if and only if the nonnegative random variables $A = [-X \mid X < 0]$ and $B = [X \mid X \geq 0]$ are q-subweibull. Let $R_{qx}$, $R_{qa}$, and $R_{qb}$ denote the radii of convergence for $X$, $A$, and $B$, respectively. Then $R_{qx} = \min\{R_{qa}, R_{qb}\}$.

**Proposition 3.1.** The following are equivalent characterizations of a q-subweibull random variable $X$ where $q > 0$.

1. Tail bound:

   (a) $\exists\, K_{1a} > 0$ such that $\forall\, t \geq 0$,

   $$\Pr(|X| > t) \leq 2 \exp\big(-(t/K_{1a})^q\big)$$

   (b) $\exists\, K_{1b} > 0$ such that

   $$\limsup_{t \to \infty} \Pr(|X| > t) \exp\big((t/K_{1b})^q\big) < \infty$$

2. Growth rate of absolute moments:

   (a) $\exists\, K_2 > 0$ such that $\forall\, p \geq 1$,

   $$\big(\mathbb{E}[|X|^p]\big)^{1/p} \leq K_2 p^{1/q}$$

   (b)

   $$\limsup_{p \to \infty} \frac{\big(\mathbb{E}[|X|^p]\big)^{1/p}}{p^{1/q}} < \infty$$

3. MGF of $|X|^q$ finite in open interval of zero: $\exists\, K_3 > 0$ such that

   (a) $\forall\, 0 < \lambda < \frac{2^{1/q}}{K_3}$

   $$\mathbb{E}\left[\exp(\lambda^q |X|^q)\right] \leq \frac{1}{1 - \lambda^q K_3^q / 2}$$

   (b) $\forall\, 0 < \lambda \leq 1/K_3$

   $$\mathbb{E}\left[\exp(\lambda^q |X|^q)\right] \leq \exp(K_3^q \lambda^q)$$

A similar result (excluding conditions 1b and 2b) was proven by Vladimirova et al. (2020). Our proof provides explicit constants, which reveals the connection with the Orlicz norm.

**Proposition 3.2.** A random variable $X$ is q-subweibull ($q \geq 1$) if and only if the $\psi_q$-Orlicz norm is finite. Furthermore, when $q \geq 1$ the constants in Proposition 3.1 are related to the norm by the global constant

$$C_q = \exp\left(\frac{2\Gamma(1/q)(eq)^{1/q}}{eq}\right)\frac{e^2}{(eq\log 2)^{1/q}} \tag{1}$$

such that

$$\frac{\|X\|_{\psi_q}}{C_q} \leq K_j \leq C_q\|X\|_{\psi_q}$$

for $K_j \in \{K_{1a}, K_2, K_3\}$

An important special case is

$$C_1 = \frac{e^3}{\log 2} \tag{2}$$

A quasinorm for $q < 1$ (heavy tails) is discussed in Kuchibhotla & Chakrabortty (2022). This relationship to the Orlicz norm allows us to restate Proposition 2.7.1(e) of Vershynin (2018) with explicit constants.

**Corollary 3.0.1.** If $X$ is a subexponential random variable with mean zero, then

$$\mathbb{E}[\exp(\lambda X)] \leq \exp(2K^2\lambda^2)$$

for all $|\lambda| \leq 1/(2K)$ where

$$K = eC_1\|X\|_{\psi_1} = \frac{e^4}{\log 2}\|X\|_{\psi_1}$$

It was shown by Vladimirova et al. (2020) that a q-subweibull distribution is also r-subweibull for all $r < q$. We now show that this also implies it is strictly r-subweibull.

**Corollary 3.0.2.** If $X$ is q-subweibull then it is strictly r-subweibull for all $r \in (0, q)$.

**Corollary 3.0.3.** Every bounded random variable is strictly q-subweibull for all $q > 0$.

*Proof.* If $X$ is bounded then there exists $M \geq 0$ such that $|X| \leq M$. Then $\mathbb{E}[\exp(\lambda^q|X|^q)] \leq \exp(\lambda^q M^q) < \infty$ for all $\lambda > 0$ and $q > 0$. □

**Corollary 3.0.4.** If $X$ is not strictly q-subweibull with $q \geq 1$ then it is not r-subweibull for any $r > q$.

### 3.1 Subweibull properties of the Poisson distribution

Corollaries 3.0.2 and 3.0.4 suggest a hierarchy of distributions based on the heaviness of the tails. Broadly q-subweibull distributions, which have a finite but nonzero radius of convergence ($R_q$), serve as "critical points" in the transition between the strictly r-subweibull regime ($r < q$), with $R_q = \infty$ and the not r-subweibull regime ($r > q$) with $R_q = 0$. However, the transition from strictly subweibull to not subweibull can be immediate, without passing through the stage of broadly subweibull. Here we provide a simple example: the Poisson tail is lighter than any exponential tail, but heavier than any weibull tail with $q > 1$.

**Proposition 3.3.** The Poisson distribution is strictly q-subweibull for $q \leq 1$ but not q-subweibull for any $q > 1$.

## 4 Concentration inequalities

Subweibull tail bounds can be straightforwardly used to improve the tightness of Bernstein's concentration inequality, which we restate here with explicit constants.

**Proposition 4.1.** If $X_1, \ldots, X_n$ are independent, subexponential random variables, then

$$\Pr\left(\left|\sum_{i=1}^n X_i\right| \geq t\right) \leq \begin{cases} 2\exp\left(\frac{-t^2}{2K^2\tilde{\sigma}^2}\right) & 0 \leq t \leq K\frac{\tilde{\sigma}^2}{\theta} \\ 2\exp\left(\frac{-t}{2K\theta}\right) & t \geq K\frac{\tilde{\sigma}^2}{\theta} \end{cases}$$

where $\theta = \max_i\{\|X_i\|_{\psi_1}\}$, $\tilde{\sigma}^2 = \sum_{i=1}^n \|X_i\|_{\psi_1}^2$ and $K = 2eC_1 = 2e^4/\log(2)$.

This is a standard result (eg, Theorem 2.8.1 of Vershynin (2018)) so the proof is omitted. We now show that if the summands have lighter than exponential tails, the bound can be tightened.

**Proposition 4.2.** *Light-tailed Bernstein inequality*

If $X_1, \ldots, X_n$ are independent, zero-mean q-subweibull random variables with $q > 1$, then

$$\Pr\left(\left|\sum_{i=1}^n X_i\right| \geq t\right) \leq \begin{cases} 2\exp\left(\frac{-t^2}{2K^2\sigma_1^2}\right) & 0 \leq t \leq K\frac{\tilde{\sigma}^2}{\theta} \\ 2\exp\left(\frac{-t^q}{C_q^q n^{q-1}\sigma_q^q}\right) & t \geq K\frac{\tilde{\sigma}_1^2}{\theta} \end{cases}$$

where $\sigma_q^q = \sum_{i=1}^n \|X_i\|_{\psi_q}^q$, $C_q$ is the global constant from Equation 1, and $\tilde{\sigma}^2, K, \theta$ are from Proposition 4.1.

For sums of heavy-tailed subweibull distributions ($q < 1$), the MGF does not exist, but it is still possible to produce a uniform tail bound.

**Proposition 4.3.** *Heavy-tailed Bernstein inequality*

If $X_1, \ldots, X_n$ are independent, zero-mean q-subweibull random variables with $q < 1$, then

$$\Pr\left(\left|\sum_{i=1}^n X_i\right| \geq t\right) \leq 2\exp\left\{\frac{-t^q}{(C_1/\log 2)\sum_{i=1}^n \||X_i|^q\|_{\psi_1}}\right\}$$

where $C_1$ is the global constant from Equation 2, and $\||X_i|^q\|_{\psi_1}$ is the subexponential norm of $|X_i|^q$.

Similar results are also found in Vladimirova et al. (2020); Kuchibhotla & Chakrabortty (2022).

## 5   Exponential tilting

**Definition 5.1.** Let $X$ be a random variable with distribution function $F$. If the BLT satisfies $\mathcal{L}_X(-\theta) < \infty$ for some $\theta \neq 0$, then the *exponentially tilted distribution* is given by

$$F_\theta(x) = \int_{-\infty}^x \frac{\exp(\theta t)}{\mathcal{L}_X(-\theta)} dF(t)$$

We adopt the convention of using $-\theta$ instead of $\theta$ so that the interpretation of the tilting parameter is consistent with other works that assume $X$ has an MGF, in which case one could equivalently require $M_X(\theta) < \infty$.

From the Radon-Nikodym theorem, $F_\theta$ is absolutely continuous with respect to $F$. Since the density function $e^{\theta x}/\mathcal{L}_X(-\theta)$ is also strictly positive, exponential tilting does not change the support. Generally speaking it is possible to produce a subexponential distribution by exponential tilting of any distribution with at least one light tail.

**Proposition 5.1.** If $X \sim F$ is a random variable having at least one light tail then exponential tilting is possible for all $\theta$ in some open interval $(-S, T)$ with $S, T \geq 0$ and $S + T > 0$. The resulting tilted distribution $F_\theta$ is subexponential with MGF $M_Z(t) = \mathcal{L}_X(-\theta - t)/\mathcal{L}_X(-\theta)$ finite for all $t \in (-S - \theta, T - \theta)$.

As an example, if $X \sim F$ is a nonnegative, heavy tailed random variable ($T = 0$), its left tail is strictly subexponential ($S = \infty$) so exponential tilting is possible for all $\theta < 0$. By Proposition 5.1 the resulting tilted distribution is subexponential and hence has lighter tails than the original distribution. On the other hand, if $X$ is broadly subexponential, exponential tilting produces another broadly subexponential distribution, with a shifted interval of convergence.

While exponential tilting can alter the tail behavior of heavy tailed and broadly subexponential distributions, it does not affect the tail behavior of q-subweibull distributions with lighter than exponential tails (i.e., $q > 1$).

**Lemma 5.0.1.** Preservation of nonnegative subweibull tails under exponential tilting. Let $\theta$ be any real number. If $X \sim F$ is nonnegative and q-subweibull ($q > 1$), then the exponentially tilted variable $Z \sim F_\theta$ is also nonnegative and q-subweibull with the same radius of convergence.

1. $\mathbb{E}[\exp(\lambda^q X^q)] < \infty$ for all $\lambda \in [0, R_q)$ implies $\mathbb{E}[\exp(\lambda^q Z^q)] < \infty$ for all $\lambda \in [0, R_q)$.

2. $\mathbb{E}[\exp(\lambda^q X^q)] = \infty$ for all $\lambda > R_q$ implies $\mathbb{E}[\exp(\lambda^q Z^q)] = \infty$ for all $\lambda > R_q$.

We now extend Lemma 5.0.1 to general random variables.

**Theorem 5.1.** Preservation of subweibull tails under exponential tilting. Let $\theta$ be any real number.

1. If $X \sim F$ is q-subweibull ($q > 1$) with radius of convergence $R_q$, then the exponentially tilted variable $Z \sim F_\theta$ is also q-subweibull and has the same radius of convergence.

2. If $X \sim F$ is strictly q-subweibull ($q \geq 1$), the exponentially tilted variable $Z \sim F_\theta$ is also strictly q-subweibull.

3. If $X \sim F$ is not q-subweibull ($q > 1$), then $Z \sim F_\theta$ is also not q-subweibull.

### 5.1 Application to domain adaptation

Consider a classification problem where labeled training data are drawn from distribution $P$ and the goal is to make accurate predictions on a different target distribution $Q$. We are given unlabeled samples from $Q$. Maity et al. (2023) proposed an exponential tilt model to facilitate importance weighting:

$$q(x, Y = k) = \exp\left\{\theta_k' T(x) + \alpha_k\right\} p(x, Y = k)$$

where $\theta_k, \alpha_k$ are the tilting parameters. It can be straightforwardly shown that

$$\alpha_k = -\log \mathbb{E}_{X \sim P}\left[\exp(\theta_k' T(X))\right] = -\log M_{T(x)}(\theta_k)$$

For simplicity we consider a univariate $T(X)$. If its distribution has heavy tails on both sides, the MGF is not finite and exponential tilting is not possible. In practice, the method of Maity et al. (2023) relies on minimizing a discrepancy such as KL divergence between $P$ and $Q$ using samples. If the distribution of $T(X)$ is heavy tailed, it may not be possible to consistently estimate the tilting parameters. If $T(X)$ has a heavy tail on only one side, then the importance weights may still be validly estimated, but the tilting parameters involved must be constrained. On the other hand, if $T(x)$ follows a q-subweibull distribution with $q > 1$, then any tail bounds available for $T(X)$ may be readily transferred to the target distribution $Q$ using Theorem 5.1.

## 6 Discussion

The theory of subexponential and subgaussian distributions is a key prerequisite to many results in theoretical machine learning and nonasymptotic statistics. That the important subexponential properties can be generalized to the broader subweibull class has been established by Vladimirova et al. (2020) and Kuchibhotla & Chakrabortty (2022). Our work differs from these in several ways: we provide explicit constants without requiring quasinorms, distinguish between strictly and broadly subweibull distributions, and address exponential tilting, which has not been previously examined to our knowledge. Exponential tilting is used in a variety of statistical areas such as causal inference (McClean et al., 2024) and Monte Carlo sampling (Fuh & Wang, 2024). If $F_\theta$ is a tilted distribution, it is a natural exponential family with parameter $\theta$. The exponential families are building blocks for generalized linear models (McCullagh & Nelder, 1989). For another application of exponential tilting to machine learning beyond domain adaptation, see Li et al. (2023).

Here, we have provided a brief overview of subweibull distributions and their Orlicz norms. We showed that the Poisson distribution is strictly 1-subweibull but not q-subweibull for any $q > 1$. We illustrated the application of subweibull properties to prove concentration inequalities. Finally, we detailed the conditions under which the subweibull property is preserved under exponential tilting. Specifically, if a distribution is subweibull with a lighter than exponential tail, then the tail of the exponentially tilted distribution decays at the same rate.

**Acknowledgments**

Thanks to Arun Kumar Kuchibhotla, Sam Power, Patrick Staples, Valerie Ventura, Larry Wasserman, and Matt Werenski for helpful comments and suggestions.

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

# A  Proofs for Section 2 (Preliminaries)

**Lemma 2.0.2.** Let $c_0, K_0 > 0$ be constants and $\psi$ an Orlicz function. The following are equivalent.

(a)
$$\mathbb{E}\left[\psi\left(\frac{|X|}{c_0}\right)\right] \leq K_0$$

(b)
$$\|X\|_\psi \leq c_0 \max\{K_0, 1\}$$

*Proof.* $(a) \implies (b)$: if $K_0 \leq 1$ the result is immediate. If $K_0 > 1$, let $\alpha \in (0, 1)$.

$$\mathbb{E}\left[\psi\left(\frac{|X|}{c_0/\alpha}\right)\right] = \mathbb{E}\left[\psi\left((1-\alpha)(0) + \alpha\frac{|X|}{c_0}\right)\right]$$
$$\leq \mathbb{E}\left[(1-\alpha)\psi(0) + \alpha\psi\left(\frac{|X|}{c_0}\right)\right]$$
$$\leq \alpha K_0$$

Setting $\alpha = 1/K_0$ produces the result.

$(b) \implies (a)$: Let $\|X\|_\psi \leq t$. Then

$$\mathbb{E}\left[\psi\left(\frac{|X|}{t}\right)\right] \leq 1$$

We need to show for every $K_1 > 0$, there exists a $c_1 > 0$ such that

$$\mathbb{E}\left[\psi\left(\frac{|X|}{c_1}\right)\right] \leq K_1$$

For all $K_1 \geq 1$ simply choose $c_1 = t$. For $K_1 < 1$,

$$\mathbb{E}\left[\psi\left(\frac{|X|}{t/K_1}\right)\right] = \mathbb{E}\left[\psi\left((1-K_1)(0) + K_1\frac{|X|}{t}\right)\right]$$
$$\leq \mathbb{E}\left[(1-K_1)\psi(0) + K_1\psi\left(\frac{|X|}{t}\right)\right]$$
$$\leq K_1$$

So we can set $c_1 \geq \max\{t, t/K_1\}$.  $\square$

# B  Proofs for Section 3 (Subweibull random variables)

**Lemma 3.0.1.** Random variable $X$ with $\Pr(X < 0) \notin \{0, 1\}$ is q-subweibull if and only if the nonnegative random variables $A = [-X \mid X < 0]$ and $B = [X \mid X \geq 0]$ are q-subweibull. Let $R_{qx}$, $R_{qa}$, and $R_{qb}$ denote the radii of convergence for $X$, $A$, and $B$, respectively. Then $R_{qx} = \min\{R_{qa}, R_{qb}\}$.

*Proof.* Let $p = \Pr(X < 0)$ and define nonnegative random variables $A = [-X \mid X < 0]$ and $B = [X \mid X \geq 0]$.

$$\mathbb{E}[\exp(\lambda^q |X|^q)] = \mathbb{E}[\exp(\lambda^q(-X)^q) \mid X < 0]p + \mathbb{E}[\exp(\lambda^q X^q) \mid X \geq 0](1-p)$$
$$= \mathbb{E}[\exp(\lambda^q A^q)]p + \mathbb{E}[\exp(\lambda^q B^q)](1-p)$$

The left hand side is finite if and only if both terms on the right hand side are finite. If $R_{qx}$ is the radius of convergence for $X$ then $\mathbb{E}[\exp(\lambda^q |X|^q)] < \infty$ for all $\lambda \in [0, R_{qx})$. Clearly $\mathbb{E}[\exp(\lambda^q A^q)] < \infty$ and $\mathbb{E}[\exp(\lambda^q B^q)] < \infty$ for all $\lambda \in [0, R_{qx})$ also, implying $\min\{R_{qa}, R_{qb}\} \geq R_{qx}$. However, if $\min\{R_{qa}, R_{qb}\} > R_{qx}$ then there exists some $\lambda > R_{qx}$ such that $\mathbb{E}[\exp(\lambda^q |X|^q)] < \infty$, which by Definition 3.2 means $R_{qx}$ is not the radius of convergence of $X$, a contradiction. Therefore, $\min\{R_{qa}, R_{qb}\} = R_{qx}$.  $\square$

**Proposition 3.1.** The following are equivalent characterizations of a q-subweibull random variable $X$ where $q > 0$.

1. Tail bound:

   (a) $\exists\, K_{1a} > 0$ such that $\forall\, t \geq 0$,

   $$\Pr(|X| > t) \leq 2 \exp\left(-(t/K_{1a})^q\right)$$

   (b) $\exists\, K_{1b} > 0$ such that

   $$\limsup_{t \to \infty} \Pr(|X| > t) \exp\left((t/K_{1b})^q\right) < \infty$$

2. Growth rate of absolute moments:

   (a) $\exists\, K_2 > 0$ such that $\forall\, p \geq 1$,

   $$\left(\mathbb{E}[|X|^p]\right)^{1/p} \leq K_2 p^{1/q}$$

   (b)

   $$\limsup_{p \to \infty} \frac{\left(\mathbb{E}[|X|^p]\right)^{1/p}}{p^{1/q}} < \infty$$

3. MGF of $|X|^q$ finite in open interval of zero: $\exists\, K_3 > 0$ such that

   (a) $\forall\, 0 < \lambda < \frac{2^{1/q}}{K_3}$

   $$\mathbb{E}\left[\exp(\lambda^q |X|^q)\right] \leq \frac{1}{1 - \lambda^q K_3^q/2}$$

   (b) $\forall\, 0 < \lambda \leq 1/K_3$

   $$\mathbb{E}\left[\exp(\lambda^q |X|^q)\right] \leq \exp(K_3^q \lambda^q)$$

*Proof.* $(1a) \implies (1b)$:

$$\limsup_{t \to \infty} \Pr(|X| > t) \exp\left((t/K_{1a})^q\right) \leq \sup_{t \geq 0} \Pr(|X| > t) \exp\left((t/K_{1a})^q\right) \leq 2 < \infty$$

So we can simply set $K_{1b} = K_{1a}$.

$(1b) \implies (1a)$: Assume

$$\limsup_{t \to \infty} \Pr(|X| > t) \exp\left((t/K_{1b})^q\right) = K$$

Then, for every $C > K$, there exists some $T$ such that for all $t > T$,

$$\Pr(|X| > t) \exp\left((t/K_{1b})^q\right) < C$$

For all $t \in [0, T]$, $\Pr(|X| > t) \leq 1$ and $\exp\left((t/K_{1b})^q\right) \leq \exp\left((T/K_{1b})^q\right)$. Therefore

$$\sup_{t \geq 0} \Pr(|X| > t) \exp\left((t/K_{1b})^q\right) \leq \max\left\{C,\ \exp\left((T/K_{1b})^q\right)\right\}$$

Let $U = \max\left\{C,\ \exp\left((T/K_{1b})^q\right)\right\}$. If $U \leq 2$ this directly implies (1a) with $K_{1a} = K_{1b}$. In the case that $U > 2$, set

$$K_{1a} = K_{1b} \left(\frac{\log U}{\log 2}\right)^{1/q} > K_{1b}$$

Let $f(t) = U \exp\left(-(t/K_{1b})^q\right)$, $g(t) = 2\exp\left(-(t/K_{1a})^q\right)$, and $T^\star = K_{1b}(\log U)^{1/q}$. Since $f(T^\star) = g(T^\star) = 1$ and $g(t)$ is a strictly decreasing function, this implies that $\Pr(|X| > t) \leq 1 \leq g(t)$ for $t \in [0, T^\star]$. For $t \geq T^\star$, $g(t) \geq f(t)$ since $K_{1a} > K_{1b}$, and $f(t) \geq \Pr(|X| > t)$ by assumption therefore $2\exp\left(-(t/K_{1a})^q\right) \geq \Pr(|X| > t)$ for all $t \geq 0$.

$(1a) \implies (2a)$:

$$\mathbb{E}[|X|^p] = \int_0^\infty \Pr(|X|^p > u)du$$

$$= \int_0^\infty \Pr(|X| > t)pt^{p-1}dt$$

$$\leq \int_0^\infty 2\exp(-(t/K_{1a})^q)pt^{p-1}dt$$

$$= (2p) \int_0^\infty \left(K_{1a}s^{1/q}\right)^{p-1} \frac{K_{1a}}{q} s^{(1/q)-1} \exp(-s)ds$$

$$= 2(p/q)K_{1a}^p \int_0^\infty s^{p/q-1}e^{-s}ds = (2p/q)K_{1a}^p\Gamma(p/q)$$

For $x \geq c > 0$, the function $\Gamma(x)$ is bounded above by $\Gamma(c)(e/c)^c(x/e)^x$. Therefore, with $c = 1/q$,

$$\Gamma(p/q) \leq \Gamma(1/q)(eq)^{1/q}(p/q)^{p/q}\exp(-p/q)$$

Substituting this into the previous expressions.

$$\mathbb{E}[|X|^p] \leq (2p/q)K_{1a}^p\Gamma(1/q)(eq)^{1/q}(p/q)^{p/q}\exp(-p/q)$$

$$\mathbb{E}[|X|^p]^{1/p} \leq \left(\frac{2\Gamma(1/q)(eq)^{1/q}}{q}p\right)^{1/p} K_{1a}(eq)^{-1/q}p^{1/q}$$

Let $a = \frac{2\Gamma(1/q)(eq)^{1/q}}{q} > 0$. Consider the function $f(x) = (ax)^{1/x}$ which we would like to upper bound.

$$f(x) = \exp\left(\frac{1}{x}\log(ax)\right)$$

$$f'(x) = f(x)\left[\frac{1}{x^2} - \frac{1}{x^2}\log(ax)\right] = \frac{f(x)}{x^2}(1 - \log(ax))$$

$$f''(x) = \frac{f'(x)}{x^2}(1 - \log(ax)) - 2\frac{f(x)}{x^3}(1 - \log(ax)) - \frac{f(x)}{x^3}$$

The global maximum occurs at $x = e/a$ which is confirmed by checking $f'(e/a) = 0$ and $f''(e/a) < 0$. Therefore a suitable upper bound is $(ax)^{1/x} \leq \exp(a/e)$. Substitute this into the above expression.

$$\mathbb{E}[|X|^p]^{1/p} \leq \exp\left(\frac{2\Gamma(1/q)(eq)^{1/q}}{eq}\right)(eq)^{-1/q}K_{1a}p^{1/q}$$

Therefore, there exists some $K_2 \leq C_2K_{1a}$ such that $\mathbb{E}[|X|^p]^{1/p} \leq K_2p^{1/q}$ as required where

$$C_2 = \exp\left(\frac{2\Gamma(1/q)(eq)^{1/q}}{eq}\right)(eq)^{-1/q}$$

Note that $C_2 > 1$ for all $q$.

$(2a) \implies (2b)$:

$$\limsup_{p\to\infty} \frac{\left(\mathbb{E}[|X|^p]\right)^{1/p}}{p^{1/q}} \leq \sup_{p\geq 1} \frac{\left(\mathbb{E}[|X|^p]\right)^{1/p}}{p^{1/q}} \leq K_2 < \infty$$

$(2b) \implies (2a)$: Assume

$$\limsup_{p\to\infty} \frac{\left(\mathbb{E}[|X|^p]\right)^{1/p}}{p^{1/q}} = K < \infty$$

Then for every $C > K$, there exists some $p^\star$ such that for all $p > p^\star$,

$$\frac{\left(\mathbb{E}[|X|^p]\right)^{1/p}}{p^{1/q}} < C$$

The $L_p$ norm is increasing in $p$, so for $p \in [1, p^\star]$, $\left(\mathbb{E}[|X|^p]\right)^{1/p} \leq \left(\mathbb{E}[|X|^{p^\star}]\right)^{1/p^\star}$ and $p^{1/q} \geq 1$, which establishes

$$\sup_{p \geq 1} \frac{\left(\mathbb{E}[|X|^p]\right)^{1/p}}{p^{1/q}} \leq \max\left\{ \left(\mathbb{E}[|X|^{p^\star}]\right)^{1/p^\star}, \ C \right\}$$

$(2a) \implies (3a)$: The power series representation of the exponential function produces

$$\mathbb{E}\left[\exp(\lambda^q |X|^q)\right] = \mathbb{E}\left[1 + \sum_{p=1}^{\infty} \frac{(\lambda^q |X|^q)^p}{p!}\right] = 1 + \sum_{p=1}^{\infty} \frac{\lambda^{pq} \mathbb{E}[|X|^{pq}]}{p!}$$

From (2) we have $\mathbb{E}[|X|^{pq}] \leq K_2^{pq}(pq)^p$ and $p! \geq (p/e)^p$ by Stirling approximation.

$$\mathbb{E}\left[\exp(\lambda^q |X|^q)\right] \leq 1 + \sum_{p=1}^{\infty} \frac{\left(\lambda^q K_2^q pq\right)^p}{(p/e)^p} = \sum_{p=0}^{\infty} \left(\lambda^q K_2^q eq\right)^p = \frac{1}{1 - \lambda^q K_2^q eq}$$

where the last series converges when $\lambda^q K_2^q eq < 1$, or

$$\lambda < \frac{1}{K_2 (eq)^{1/q}} = \frac{2^{1/q}}{K_2 (2eq)^{1/q}}$$

Since $(2eq)^{1/q} \leq e^2$ for all $q > 0$, and $1/(1-x)$ is increasing for $x < 1$, set $C_3 = e^2$ so that $\lambda < \frac{2^{1/q}}{C_3 K_2}$ implies series convergence and

$$\mathbb{E}\left[\exp(\lambda^q |X|^q)\right] \leq \frac{1}{1 - \lambda^q K_2^q eq} = \frac{1}{1 - \lambda^q \left((2eq)^{1/q} K_2\right)^q / 2} \leq \frac{1}{1 - \lambda^q (C_3 K_2)^q / 2}$$

Therefore there exists some $K_3 \leq C_3 K_2$ such that if $\lambda < \frac{2^{1/q}}{K_3}$,

$$\mathbb{E}\left[\exp(\lambda^q |X|^q)\right] \leq \frac{1}{1 - \lambda^q K_3^q / 2}$$

Clearly $C_3 > 1$ for all $q$ as well.

$(3a) \implies (3b)$: Straightforward application of the numerical inequality $\frac{1}{1-x} \leq e^{2x}$ for $0 \leq x \leq 1/2$.

$(3b) \implies (1a)$: Set $C_1 = (\log 2)^{-1/q}$ and note that $C_1 > 1$ for all $q > 0$. Therefore we may choose

$$\lambda = \frac{(\log 2)^{1/q}}{K_3} = \frac{1}{C_1 K_3} \leq \frac{1}{K_3}$$

so that $\mathbb{E}[\exp(\lambda^q |X|^q)] \leq \exp(\lambda^q K_3^q) = 2$. Then,

$$\begin{aligned}
\Pr(|X| > t) &= \Pr\left(\exp(\lambda^q |X|^q) > \exp(\lambda^q t^q)\right) \\
&\leq \mathbb{E}[\exp(\lambda |X|^q)] \exp(-(\lambda t)^q) \\
&\leq 2 \exp\left(-\left(\frac{t}{C_1 K_3}\right)^q\right)
\end{aligned}$$

So there exists some $K_{1a} \leq C_1 K_3$ such that condition (1a) is satisfied as desired. $\qquad \square$

**Proposition 3.2.** A random variable $X$ is q-subweibull ($q \geq 1$) if and only if the $\psi_q$-Orlicz norm is finite. Furthermore, when $q \geq 1$ the constants in Proposition 3.1 are related to the norm by the global constant

$$C_q = \exp\left(\frac{2\Gamma(1/q)(eq)^{1/q}}{eq}\right)\frac{e^2}{(eq\log 2)^{1/q}} \tag{1}$$

such that

$$\frac{\|X\|_{\psi_q}}{C_q} \leq K_j \leq C_q\|X\|_{\psi_q}$$

for $K_j \in \{K_{1a}, K_2, K_3\}$

*Proof.* Proposition 3.1 (3b) implies finite norm: Let $\psi_q(x) = \exp(x^q) - 1$ as before. This is a convex function for $q \geq 1$. Rearranging terms produces

$$\mathbb{E}\left[\psi_q\left(\frac{|X|}{1/\lambda}\right)\right] = \mathbb{E}\left[\exp(\lambda^q|X|^q)\right] - 1 \leq \exp(\lambda^q K_3^q) - 1$$

From Lemma 2.0.2 this implies

$$\|X\|_{\psi_q} \leq (1/\lambda)\max\left\{1,\ \exp(K_3^q\lambda^q) - 1\right\}$$

where $0 < \lambda < 1/K_3$. Observing that the second term in the max is increasing in $\lambda$ and ranges from 0 to $e - 1 > 1$, while $1/\lambda$ is a decreasing function, the tightest bound in terms of $\lambda$ is achieved when $\exp(K_3^q\lambda^q) - 1 = 1$ which produces $\lambda^\star = \left(\log 2\right)^{1/q}/K_3$. This establishes $\|X\|_{\psi_q} \leq C_4 K_3 < \infty$ with $C_4 = (\log 2)^{-1/q}$.

Finite norm implies Proposition 3.1 (1a): Let $K_4 = \|X\|_{\psi_q}(1 + \epsilon)$ for some arbitrarily small $\epsilon > 0$.

$$\begin{aligned}
\Pr(|X| \geq t) &= \Pr\left(\exp\left(\left(\frac{|X|}{K_4}\right)^q\right) \geq \exp\left((t/K_4)^q\right)\right) \\
&\leq \mathbb{E}\left[\exp\left(\left(\frac{|X|}{K_4}\right)^q\right)\right]\exp\left(-(t/K_4)^q\right) \\
&\leq 2\exp\left(-(t/K_4)^q\right)
\end{aligned}$$

This shows there exists some $K_{1a} \leq K_4 = \|X\|_{\psi_q}(1 + \epsilon)$ such that the desired condition is satisfied.

To show the relationship to the global constant, recall from the proof of Proposition 3.1 that

$$\begin{aligned}
C_1 &= C_4 = (\log 2)^{-1/q} \\
C_2 &= \exp\left(\frac{2\Gamma(1/q)(eq)^{1/q}}{eq}\right)(eq)^{-1/q} \\
C_3 &= e^2
\end{aligned}$$

Set

$$C_q = C_1 C_2 C_3 = \exp\left(\frac{2\Gamma(1/q)(eq)^{1/q}}{eq}\right)\frac{e^2}{(eq\log 2)^{1/q}}$$

Since $C_1, C_2, C_3 \geq 1$, $C_q \geq 1$ also. We can choose $\epsilon \leq C_1 - 1 > 0$ so that

$$K_{1a} \leq \|X\|_{\psi_q}(1 + \epsilon) \leq C_1\|X\|_{\psi_q} \leq C_q\|X\|_{\psi_q}$$

and

$$\|X\|_{\psi_q} \leq C_1 K_3 \leq C_1 C_3 K_2 \leq C_1 C_3 C_2 K_{1a} = C_q K_{1a}$$

By a similar argument,

$$K_2 \leq C_2 K_{1a} \leq C_2 C_1\|X\|_{\psi_q} \leq C_q\|X\|_{\psi_q}$$

and

$$\|X\|_{\psi_q} \leq C_1 K_3 \leq C_1 C_3 K_2 \leq C_q K_2$$

Finally,

$$K_3 \leq C_3 C_2 K_{1a} \leq C_3 C_2 C_1 \|X\|_{\psi_q} = C_q \|X\|_{\psi_q}$$

and $\|X\|_{\psi_q} \leq C_1 K_3 \leq C_q K_3$. □

**Corollary 3.0.1.** If $X$ is a subexponential random variable with mean zero, then

$$\mathbb{E}[\exp(\lambda X)] \leq \exp(2K^2 \lambda^2)$$

for all $|\lambda| \leq 1/(2K)$ where

$$K = eC_1 \|X\|_{\psi_1} = \frac{e^4}{\log 2} \|X\|_{\psi_1}$$

*Proof.* By Propositions 3.1 and 3.2 there is some $K_2 \leq C_1 \|X\|_{\psi_1}$ such that $\mathbb{E}[|X|^p] \leq K_2^p p^p$ for all $p \geq 1$ where $C_1 = e^3/\log(2)$ (Equation 2).

$$\mathbb{E}[\exp(\lambda x)] = 1 + (0) + \sum_{p=2}^{\infty} \frac{\lambda^p \mathbb{E}[X^p]}{p!} \leq 1 + \sum_{p=2}^{\infty} \frac{|\lambda|^p K_2^p p^p}{(p/e)^p} = 1 + \frac{(\lambda e K_2)^2}{1 - |\lambda| e K_2}$$

where the sum converges whenever $|\lambda e K_2| < 1$. The function $1 + x^2/(1 - |x|)$ is bounded above by $\exp(2x^2)$ whenever $|x| \leq 1/2$, so

$$\mathbb{E}[\exp(\lambda x)] \leq \exp\left(2(\lambda e K_2)^2\right) \leq \exp\left(2\lambda^2 \left(eC_1 \|X\|_{\psi_1}\right)^2\right)$$

whenever $|\lambda e K_2| \leq 1/2$ which is satisfied by

$$|\lambda| \leq \frac{1}{2eC_1 \|X\|_{\psi_1}}$$

Setting $K = eC_1 \|X\|_{\psi_1}$ yields the desired result. □

**Corollary 3.0.2.** If $X$ is q-subweibull then it is strictly r-subweibull for all $r \in (0, q)$.

*Proof.* If $X$ is q-subweibull, by Proposition 3.1 we may assume there exists $K > 0$ such that $\forall\, p \geq 1$,

$$\left(\mathbb{E}[|X|^p]\right)^{1/p} \leq K p^{1/q}$$

Let $r \in (0, q)$. The MGF of $|X|^r$ is given by

$$\mathbb{E}\left[\exp(\lambda^r |X|^r)\right] = 1 + \sum_{p=1}^{\infty} \frac{\lambda^{pr} E[|X|^{pr}]}{p!}$$

$$\leq 1 + \sum_{p=1}^{\infty} \frac{\lambda^{pr} K^{pr} (pr)^{pr/q}}{(p/e)^p}$$

$$= 1 + \sum_{p=1}^{\infty} \left(\lambda^r K^r e r^{r/q}\right)^p p^{p(r/q-1)}$$

Apply the root test to the series to determine convergence.

$$R(p) = \lambda^r K^r e r^{r/q} p^{r/q-1}$$

Since $r < q$, then $\lim_{p \to \infty} R(p) = 0$ and the series converges regardless of the value of $\lambda$, which shows $X$ is strictly r-subweibull. □

**Corollary 3.0.4.** If $X$ is not strictly q-subweibull with $q \geq 1$ then it is not r-subweibull for any $r > q$.

*Proof.* From Proposition 3.1 we may assume $\exists \, \lambda > 0$ such that

$$\limsup_{t \to \infty} \Pr(|X| > t) \exp(\lambda t^q) = \infty$$

which implies there is an infinite sequence $t_n \to \infty$ such that

$$\lim_{n \to \infty} \Pr(|X| > t_n) \exp(\lambda t_n^q) = \infty$$

Now let $\rho > 0$ and $r > q$. Whenever $t \geq t^\star = (\lambda/\rho)^{1/(r-q)}$, we have $\exp(\rho t^r) \geq \exp(\lambda t^q)$. Let $\{t_m\}$ be the infinite subsequence of $\{t_n\}$ excluding the elements less than $t^\star$. Clearly $t_m \to \infty$ as well. Then

$$\lim_{m \to \infty} \Pr(|X| > t_m) \exp(\rho t_m^r) \geq \lim_{m \to \infty} \Pr(|X| > t_m) \exp(\lambda t_m^q) = \infty$$

which implies $X$ cannot be r-subweibull. $\qquad \square$

**Proposition 3.3.** The Poisson distribution is strictly q-subweibull for $q \leq 1$ but not q-subweibull for any $q > 1$.

*Proof.* Since the Poisson distribution has a finite MGF with infinite radius of convergence, it is strictly subexponential and by Corollary 3.0.2 strictly q-subweibull for all $q \leq 1$. Let $X \sim Poi(\mu)$. Without loss of generality assume $t > 1$ and let $n = \lfloor t \rfloor + 1$ with $t < n \leq t + 1$.

$$\Pr(X > t) = \sum_{j=n}^{\infty} \Pr(X = j) \geq \Pr(X = n) = \frac{\mu^n \exp(-\mu)}{n!} = \frac{\mu^n \exp(-\mu)}{n\Gamma(n)}$$

Since $t\Gamma(t)$ is increasing for $t \geq 1$, we have $n\Gamma(n) \leq (t+1)\Gamma(t+1)$. Also, $\Gamma(n) \leq n^n$ for $n \geq 1$. For the $\mu^n$ term, it is increasing for $\mu \geq 1$ and decreasing for $\mu < 1$, so $\mu^n \geq \min\{\mu^{t+1}, \mu^t\} = \mu^t \min\{\mu, 1\}$. Combining these we obtain

$$\Pr(X > t) \geq \frac{\mu^t \min\{\mu, 1\} e^{-\mu}}{(t+1)\Gamma(t+1)} = \frac{\mu^t \min\{\mu, 1\} e^{-\mu}}{(t+1)(t)\Gamma(t)} \geq \frac{\mu^t \min\{\mu, 1\} e^{-\mu}}{(t+1)(t)t^t}$$

To assess whether the tail follows a subweibull rate of decay, choose any $\lambda > 0$ and $q > 1$, then

$$\limsup_{t \to \infty} \Pr(X > t) \exp(\lambda t^q) \geq \min\{\mu, 1\} e^{-\mu} \lim_{t \to \infty} \frac{\mu^t}{(t+1)(t)t^t} \exp(\lambda t^q)$$

$$= \min\{\mu, 1\} e^{-\mu} \exp\left[\lim_{t \to \infty} t \log \mu - \log(t+1) - \log t - t \log t + \lambda t^q\right]$$

The expression inside brackets is of the form $\infty - \infty$ so we rearrange terms and apply L'Hopital's rule. Define

$$\lim_{t \to \infty} t \log \mu - \log(t+1) - \log t - t \log t + \lambda t^q$$

$$= \lim_{t \to \infty} (t \log t) \left[\lim_{t \to \infty} \frac{\log \mu}{\log t} - \frac{\log(t+1)}{t \log t} - \frac{1}{t} + \frac{\lambda t^q}{t \log t}\right]$$

$$= \lim_{t \to \infty} (t \log t) \left[\lim_{t \to \infty} 0 - \frac{1/(t+1)}{1 + \log t} - (0) + \frac{\lambda q t^{q-1}}{1 + \log t}\right]$$

$$= \lim_{t \to \infty} (t \log t) \left[\lim_{t \to \infty} \frac{\lambda q (q-1) t^{q-2}}{1/t}\right] = \infty \cdot \infty = \infty$$

Therefore

$$\limsup_{t \to \infty} \Pr(X > t) \exp(\lambda t^q) = \infty$$

Since this holds for all $\lambda > 0$, $X$ cannot satisfy Proposition 3.1 and therefore is not q-subweibull for any $q > 1$. $\qquad \square$

## C   Proofs for Section 4 (Concentration inequalities)

**Proposition 4.2.** *Light-tailed Bernstein inequality*

If $X_1, \ldots, X_n$ are independent, zero-mean q-subweibull random variables with $q > 1$, then

$$\Pr\left(\left|\sum_{i=1}^{n} X_i\right| \geq t\right) \leq \begin{cases} 2\exp\left(\frac{-t^2}{2K^2\sigma_1^2}\right) & 0 \leq t \leq K\frac{\tilde{\sigma}^2}{\theta} \\ 2\exp\left(\frac{-t^q}{C_q^q n^{q-1}\sigma_q^q}\right) & t \geq K\frac{\tilde{\sigma}_1^2}{\theta} \end{cases}$$

where $\sigma_q^q = \sum_{i=1}^{n} \|X_i\|_{\psi_q}^q$, $C_q$ is the global constant from Equation 1, and $\tilde{\sigma}^2, K, \theta$ are from Proposition 4.1.

*Proof.* By Corollary 3.0.2 each $X_i$ is subexponential, so the bound for small deviations $t$ follows directly from Proposition 4.1. For large deviations, let $S = \sum_{i=1}^{n} X_i$. Since $q > 1$ the Orlicz norm exists and by subadditivity,

$$\|S\|_{\psi_q} = \left\|\sum_{i=1}^{n} X_i\right\|_{\psi_q} \leq \sum_{i=1}^{n} \|X_i\|_{\psi_q} \leq n^{1-1/q}\left(\sum_{i=1}^{n} \|X_i\|_{\psi_q}^q\right)^{1/q} = n^{1-1/q}\sigma_q$$

By Propositions 3.1 and 3.2, there exists some $K_{1a} \leq C_q\|S\|_{\psi_q}$ such that

$$\Pr(|S| \geq t) \leq 2\exp(-(t/K_{1a})^q) \leq 2\exp\left(\frac{-t^q}{C_q^q n^{q-1}\sigma_q^q}\right)$$

For all $t \geq 0$. In particular, it holds for $t \geq K\tilde{\sigma}^2/\theta$. $\qquad\square$

**Proposition 4.3.** *Heavy-tailed Bernstein inequality*

If $X_1, \ldots, X_n$ are independent, zero-mean q-subweibull random variables with $q < 1$, then

$$\Pr\left(\left|\sum_{i=1}^{n} X_i\right| \geq t\right) \leq 2\exp\left\{\frac{-t^q}{(C_1/\log 2)\sum_{i=1}^{n} \||X_i|^q\|_{\psi_1}}\right\}$$

where $C_1$ is the global constant from Equation 2, and $\||X_i|^q\|_{\psi_1}$ is the subexponential norm of $|X_i|^q$.

*Proof.* Since $f(x) = x^q$ is concave for $x \geq 0$ and $q \in (0,1)$,

$$\left|\sum_{i=1}^{n} X_i\right|^q \leq \left(\sum_{i=1}^{n} |X_i|\right)^q \leq \sum_{i=1}^{n} |X_i|^q$$

With $\lambda > 0$ this implies

$$\Pr\left(\left|\sum_{i=1}^{n} X_i\right| \geq t\right) \leq \Pr\left(\sum_{i=1}^{n} |X_i|^q \geq t^q\right) \leq \prod_{i=1}^{n} \mathbb{E}\left[\exp\left\{\lambda|X_i|^q\right\}\right]\exp\left\{-\lambda t^q\right\}$$

If $X_i$ is q-subweibull, then $|X_i|^q$ is subexponential. By Proposition 3.1 (3b), for each $X_i$ there is some $K_{3i} \leq C_1\||X_i|^q\|_{\psi_1}$ such that

$$\mathbb{E}[\exp(\lambda|X_i|^q)] \leq \exp(K_{3i}\lambda)$$

if $\lambda \leq 1/K_{3i}$. Therefore set $\theta_q = \max_i\{\||X_i|^q\|_{\psi_1}\}$ and restrict $\lambda \leq 1/(C_1\theta_q)$. Plugging this back into the previous expression,

$$\Pr\left(\left|\sum_{i=1}^{n} X_i\right| \geq t\right) \leq \exp\left(\sum_{i=1}^{n} K_{3i}\lambda - \lambda t^q\right) \leq \exp\left(\lambda(C_1\sigma_q - t^q)\right)$$

where $\sigma_q = \sum_{i=1}^n \| |X_i|^q \|_{\psi_1}$. The bound is decreasing in $\lambda$ when $t \geq t^\star = (C_1 \sigma_q)^{1/q}$, in which case the tightest bound occurs with $\lambda = 1/(C_1 \theta_q)$. This leads to

$$\Pr\left(\left|\sum_{i=1}^n X_i\right| \geq t\right) \leq \exp\left(\frac{1}{C_1\theta_q}(C_1\sigma_q - t^q)\right) = \exp\left(\frac{\sigma_q}{\theta_q} - \frac{t^q}{C_1\theta_q}\right)$$

Referring to Proposition 3.1 (1b), we can see that

$$\limsup_{t\to\infty} \exp\left(\frac{\sigma_q}{\theta_q} - \frac{t^q}{C_1\theta_q}\right)\exp\left((t/K_{1b})^q\right) < \infty$$

with $K_{1b} = (C_1\theta_q)^{1/q}$. Therefore the sum of q-subweibull random variables is also q-subweibull. To get a bound over all $t \geq 0$, define

$$f(t) = \frac{\sigma_q}{\theta_q} - (t/K_{1b})^q$$
$$g(t) = \log(2) - (t/K_{1a})^q$$

Since $\sigma_q/\theta_q \geq 1 > \log(2)$, we can choose $K_{1a} > K_{1b}$ such that $f(t) \geq g(t) \geq 0$ for $t \leq t^\star$ and $g(t) \geq f(t)$ for $t \geq t^\star$, where $f(t^\star) = g(t^\star) = 0$.

$$t^\star = \left(\frac{\sigma_q}{\theta_q}\right)^{1/q} K_{1b} = (\log 2)^{1/q} K_{1a}$$

therefore

$$K_{1a} = \left(\frac{\sigma_q}{\theta_q \log 2}\right)^{1/q} (C_1\theta_q)^{1/q} = \left(\frac{C_1\sigma_q}{\log 2}\right)^{1/q}$$

$\square$

## D   Proofs for Section 5 (Exponential tilting)

**Proposition 5.1.** If $X \sim F$ is a random variable having at least one light tail then exponential tilting is possible for all $\theta$ in some open interval $(-S, T)$ with $S, T \geq 0$ and $S + T > 0$. The resulting tilted distribution $F_\theta$ is subexponential with MGF $M_Z(t) = \mathcal{L}_X(-\theta - t)/\mathcal{L}_X(-\theta)$ finite for all $t \in (-S - \theta, T - \theta)$.

*Proof.* Without loss of generality assume the right tail is light so $\mathcal{L}_X(-\theta) < \infty$ for some $\theta > 0$. For all $\theta' \in [0, \theta)$,

$$\mathcal{L}_X(-\theta') = \mathbb{E}[\exp(\theta' X)] \leq \mathbb{E}[\exp(\theta X)] < \infty$$

Set $T = \sup\{\theta : \mathcal{L}_X(-\theta) < \infty\} > 0$. If $X$ has a heavy left tail then $\mathcal{L}_X(-\theta) = \infty$ for all $\theta < 0$, so the interval of convergence is $(-S, T)$ with $S = 0$. If $X$ has a light left tail then we can set $S = -\inf\{\theta : \mathcal{L}_X(-\theta) < \infty\} > 0$. This establishes the interval is $(-S, T)$ with $S, T \geq 0$ and $S + T > 0$. Let $Z \sim F_\theta$ follow the tilted distribution with $\theta \in (-S, T)$. Its BLT is

$$\mathcal{L}_Z(t) = \mathbb{E}[\exp(-tZ)] = \int_{-\infty}^{\infty} \exp(-tz) dF_\theta(z) = \int_{-\infty}^{\infty} \exp(-tx)\frac{\exp(\theta x)}{\mathcal{L}_X(-\theta)} dF(x)$$
$$= \mathbb{E}[\exp(-(t-\theta)X)]/\mathcal{L}_X(-\theta) = \mathcal{L}_X(-(\theta - t))/\mathcal{L}_X(-\theta)$$

This is finite when $\theta - t \in (-S, T)$ or equivalently $t \in (-T + \theta, S + \theta)$. Since $\theta \in (-S, T)$, the interval of convergence for $\mathcal{L}_Z(t)$ is an open interval containing zero, which proves $Z$ is subexponential and has the MGF

$$M_Z(t) = \mathcal{L}_Z(-t) = \mathcal{L}_X(-\theta - t)/\mathcal{L}_X(-\theta)$$

which is finite on the interval $t \in (-S - \theta, T - \theta)$.

$\square$

**Lemma 5.0.1.** Preservation of nonnegative subweibull tails under exponential tilting. Let $\theta$ be any real number. If $X \sim F$ is nonnegative and q-subweibull $(q > 1)$, then the exponentially tilted variable $Z \sim F_\theta$ is also nonnegative and q-subweibull with the same radius of convergence.

1. $\mathbb{E}[\exp(\lambda^q X^q)] < \infty$ for all $\lambda \in [0, R_q)$ implies $\mathbb{E}[\exp(\lambda^q Z^q)] < \infty$ for all $\lambda \in [0, R_q)$.

2. $\mathbb{E}[\exp(\lambda^q X^q)] = \infty$ for all $\lambda > R_q$ implies $\mathbb{E}[\exp(\lambda^q Z^q)] = \infty$ for all $\lambda > R_q$.

*Proof.* If $X$ is q-subweibull with $q > 1$ then by Corollary 3.0.2 it is strictly subexponential and $\mathcal{L}_X(-\theta) < \infty$ for all $\theta \in \mathbb{R}$. Let $Z \sim F_\theta$. The MGF of $Z^q$ is

$$\mathbb{E}[\exp(\lambda^q Z^q)] = \frac{\int \exp(\lambda^q x^q + \theta x) dF(x)}{\mathcal{L}_X(-\theta)}$$

(1) case of $\lambda < R_q$. If $\theta \leq 0$ then

$$\int \exp(\lambda^q x^q + \theta x) dF(x) \leq \int \exp(\lambda^q x^q + 0) dF(x) = \mathbb{E}[\exp(\lambda^q X^q)] < \infty$$

If $\theta > 0$, choose $\rho \in (\lambda, R_q)$ and define

$$x^\star = \left(\frac{\theta}{\rho^q - \lambda^q}\right)^{\frac{1}{q-1}}$$

Then for $x > x^\star$, $\lambda^q x^q + \theta x \leq \rho^q x^q$. Therefore,

$$\begin{aligned}
\int \exp(\lambda^q x^q + \theta x) dF(x) &= \int_0^{x^\star} \exp(\lambda^q x^q + \theta x) dF(x) + \int_{x^\star}^\infty \exp(\lambda^q x^q + \theta x) dF(x) \\
&\leq \int_0^{x^\star} \exp\left(\theta x^\star + \lambda^q (x^\star)^q\right) dF(x) + \int_{x^\star}^\infty \exp(\rho^q x^q) dF(x) \\
&\leq \exp\left(\theta x^\star + \lambda^q (x^\star)^q\right) \Pr(X \leq x^\star) + \int_0^\infty \exp(\rho^q x^q) dF(x) \\
&< \infty
\end{aligned}$$

(2) case of $\lambda > R_q$. If $\theta \geq 0$ then

$$\int \exp(\lambda^q x^q + \theta x) dF(x) \geq \int \exp(\lambda^q x^q + 0) dF(x) = \mathbb{E}[\exp(\lambda^q X^q)] = \infty$$

If $\theta < 0$. Choose $\rho \in (R_q, \lambda)$ and define

$$x^\star = \left(\frac{-\theta}{\lambda^q - \rho^q}\right)^{\frac{1}{q-1}}$$

Then for $x > x^\star$, $\lambda^q x^q + \theta x \geq \rho^q x^q$. Therefore,

$$\begin{aligned}
\int \exp(\lambda^q x^q + \theta x) dF(x) &= \int_0^{x^\star} \exp(\lambda^q x^q + \theta x) dF(x) + \int_{x^\star}^\infty \exp(\lambda^q x^q + \theta x) dF(x) \\
&\geq \int_0^{x^\star} \exp(\lambda^q x^q + \theta x) dF(x) + \int_{x^\star}^\infty \exp(\rho^q x^q) dF(x)
\end{aligned}$$

The first term is finite. We will show the second term is infinite. By assumption,

$$\int_0^\infty \exp(\rho^q x^q) dF(x) = \infty$$

$$= \int_0^{x^\star} \exp(\rho^q x^q) dF(x) + \int_{x^\star}^\infty \exp(\rho^q x^q) dF(x)$$

But

$$\int_0^{x^\star} \exp(\rho^q x^q) dF(x) \leq \exp\left(\rho^q (x^\star)^q\right) \Pr(X \leq x^\star) < \infty$$

Therefore

$$\int_{x^\star}^{\infty} \exp(\rho^q x^q) dF(x) = \infty$$

implying

$$\int \exp(\lambda^q x^q + \theta x) dF(x) = \infty$$

as well. □

**Theorem 5.1.** Preservation of subweibull tails under exponential tilting. Let $\theta$ be any real number.

1. If $X \sim F$ is q-subweibull ($q > 1$) with radius of convergence $R_q$, then the exponentially tilted variable $Z \sim F_\theta$ is also q-subweibull and has the same radius of convergence.

2. If $X \sim F$ is strictly q-subweibull ($q \geq 1$), the exponentially tilted variable $Z \sim F_\theta$ is also strictly q-subweibull.

3. If $X \sim F$ is not q-subweibull ($q > 1$), then $Z \sim F_\theta$ is also not q-subweibull.

*Proof.* (1) By Corollary 3.0.2, $X$ is strictly subexponential so $\mathcal{L}_X(-\theta) < \infty$ for all $\theta \in \mathbb{R}$. Choose any arbitrary $\theta$ and set $M_1 = \mathcal{L}_X(-\theta)$. Define nonnegative random variables $A = [-X \mid X < 0]$ and $B = [X \mid X \geq 0]$ with distributions $F^-$ and $F^+$, respectively. By Lemma 3.0.1 both $A$ and $B$ are q-subweibull and strictly subexponential. Let $R_{qa}$ and $R_{qb}$ be the radii of convergence of $A$ and $B$, respectively. Let $p = \Pr(X < 0)$ and assume $p \notin \{0, 1\}$ (otherwise simply apply Lemma 5.0.1 to $X$ or $-X$). Note that

$$M_1 = \mathcal{L}_A(\theta) p + \mathcal{L}_B(-\theta)(1 - p) \tag{3}$$

We have

$$\mathbb{E}[\exp(\lambda^q |Z|^q)] = \int_{-\infty}^{\infty} \exp(\lambda^q |z|^q) dF_\theta(z) = \int_{-\infty}^{\infty} \exp(\lambda^q |x|^q) \frac{\exp(\theta x)}{M_1} dF(x)$$

$$= \frac{\mathbb{E}[\exp(\lambda^q |X|^q + \theta X)]}{M_1}$$

$$= \frac{p}{M_1} \mathbb{E}[\exp(\lambda^q A^q - \theta A)] + \frac{(1 - p)}{M_1} \mathbb{E}[\exp(\lambda^q B^q + \theta B)]$$

$$= \frac{p \mathcal{L}_A(\theta)}{M_1} \int_0^{\infty} \exp(\lambda^q x^q) \frac{\exp(-\theta x)}{\mathcal{L}_A(\theta)} dF^-(x) \dots$$

$$\dots + \frac{(1 - p) \mathcal{L}_B(-\theta)}{\mathcal{L}_X(-\theta)} \int_0^{\infty} \exp(\lambda^q x^q) \frac{\exp(\theta x)}{\mathcal{L}_B(-\theta)} dF^+(x)$$

$$= (\tilde{p}) \int_0^{\infty} \exp(\lambda^q z^q) dF_{(-\theta)}^-(z) + (1 - \tilde{p}) \int_0^{\infty} \exp(\lambda^q z^q) dF_\theta^+(z)$$

$$= \tilde{p} \mathbb{E}[\exp(\lambda^q U^q)] + (1 - \tilde{p}) \mathbb{E}[\exp(\lambda^q V^q)]$$

where (see Equation 3), $\tilde{p} = p \mathcal{L}_A(\theta) / M_1$ so that $\tilde{p} \in (0, 1)$. The nonnegative random variable $U$ is distributed as $F_{(-\theta)}^-$, which is the exponentially tilting of $A \sim F^-$ by $-\theta$ and $V \sim F_\theta^+$ is similarly defined as the exponential tilting of $B \sim F^+$. By Lemma 5.0.1, this implies $U$ and $V$ are q-subweibull with radii of convergence $R_{qa}$ and $R_{qb}$, respectively. Let $R_{qz}$ be the radius of convergence of $Z$. Note that

$$\Pr(Z \geq 0) = \int_0^{\infty} dF_\theta(z) = \int_0^{\infty} \frac{\exp(\theta x)}{\mathcal{L}_X(-\theta)} dF(x) = \frac{\mathbb{E}[\exp(\theta X) \mid X \geq 0] \Pr(X \geq 0)}{M_1}$$

$$= \frac{\mathbb{E}[\exp(\theta B)](1 - p)}{M_1} = \frac{\mathcal{L}_B(-\theta)(1 - p)}{M_1}$$

$$= 1 - \tilde{p}$$

So $\Pr(Z < 0) = \tilde{p}$. By Lemma 3.0.1 this implies $Z$ is q-subweibull with radius of convergence $\min\{R_{qa}, R_{qb}\}$, which is also the radius of convergence of $X$.

(2) For $q = 1$ apply Proposition 5.1 with $S = \infty$ and $T = \infty$. For $q > 1$, apply (1) with $R_q = \infty$.

(3) This is a direct corollary of (1) obtained in the case of $R_q = 0$. $\qquad\square$

