# OpenReview forum: "Exponential tilting of subweibull distributions"
_TMLR — Accepted by TMLR_

### Review · Reviewer_sxKF · 2025-07-21

**Summary Of Contributions:**

The authors studied the properties of sub-Weibull distributions, a class of distributions that interpolates between sub-exponential and sub-Gaussian. In particular, exponentially tilting preserves three separate senses of sub-Weibull distributions. The contributions are mainly technical, without direct discussions of applications, but I believe this is a great fit for TMLR. This type of technical articles will become greatly useful when one searches for the exact type of concentration results needed for the application, and therefore deserve to be published.

**Additional Comments:**

N/A

**Audience:**

Yes

**Audience Explanation:**

Concentration inequalities in general are ubiquitous in ML theory, and this particularly nice class of interpolation between sub-exponential and sub-Gaussian is very interesting on its own. Certainly any additional properties of exponential tilting will prove to be interesting as well.

**Claims And Evidence:**

Yes

**Claims Explanation:**

The authors provided clear and concise proof of the main results in the appendix.

**Requested Changes:**

N/A

---

### Review · Reviewer_q7uH · 2025-07-22

**Summary Of Contributions:**

The paper surveys the class of q-sub-Weibull distributions as natural generalizations of sub-exponential and sub-Gaussian distributions. Among others, the authors show some monotonicity properties with respect to q, the parameter determining the tails, and preservation under tilting.

**Audience:**

Yes

**Audience Explanation:**

The so-called q-sub-Weibull distributions have attracted some attention recently in various contexts, especially since they can model some heavier-tailed distributions. I believe a survey on these distributions can be of service to the community.

**Claims And Evidence:**

Yes

**Claims Explanation:**

All claims come equipped with proofs, which I have verified to be true.

**Requested Changes:**

As I indicated above, a survey on q-sub-Weibull distributions could be a useful resource. However, in its current form, I am not sure how to classify the main purpose of the paper. On the one hand, it contains a short list of results on q-sub-Weibull distributions, which seems too lean to be called a survey. On the other hand, I don't completely see the motivation for the specific results that are proven and stated in the paper. Without understanding the purpose of the paper, I will not be able to recommend its acceptance.

I have two main suggestions for the paper, which should be understood as a *broad restructuring of the paper*:

1. If it is indeed intended to be a survey, please survey the wealth of possible results on q-sub-Weibull.  Just as an example (only one among many), one could prove concentration inequalities a la Bernstein, and there is already a big body of relevant literature on Orlicz spaces. Another point would be to discuss applications and appearances in Statistics or other relevant fields.

2. Please expand on the results that already appear in the paper. If the authors had a specific application for proving, say, Theorem 4.1, then that application should appear, or at least be described in the paper. The same thing is also true for the results appearing in Section 3.

---

> ### Comment · Reviewer_q7uH · 2025-08-26
> **Response to revision**
>
> I have now read the new revision. I feel like the additional result makes the paper somewhat more appealing now, and I thank the authors for these changes.
> The only thing that is missing in my eyes now is a serious discussion about the motivation for writing the paper and introducing these results, especially about tilting. In other words, I think the paper currently misses a crucial point that explains why it is of interest to the wider TMLR audience. I would be happy to recommend acceptance after this is taken care of.

---

### Review · Reviewer_ghv5 · 2025-08-03

**Summary Of Contributions:**

1. The paper provides a concise, well‐organized summary of known results on $q$-subweibull distribution: tail‐decay bounds, moment‐growth rates, and Laplace‐transform characterizations (originally due to Vladimirova et al. 2020; Kuchibhotla & Chakrabortty 2022)—and frames them in a unified “strict vs. broad” radius‐of‐convergence perspective.

2. By showing the Poisson law is strictly 1-subweibull but not $q$-subweibull for any $q>1$, the authors give a clean, concrete example of how the subweibull classes transition from “light” to “too heavy.”

3. The main contribution is the result that exponential tilting preserves the $q$-subweibull property (strict or broad) and **exactly** the same radius $R_q$ for all $q>1$.  While tilt-invariance is well known for subgaussian/subexponential cases, this result is extended to a more general class.

**Additional Comments:**

N/A

**Audience:**

Yes

**Audience Explanation:**

Yes, the results would be of interest to people who use tilted measures and concentration inequalities for distributions in the q-subweibull class.

**Claims And Evidence:**

Yes

**Claims Explanation:**

Definitions, lemmas, and proofs are concise and well-written, and organized.

**Requested Changes:**

* The manuscript in its current form reads more like an advanced coursework exercise than a full research paper. Not sure if it meets TMLR's expectations, but the authors can consider augmenting Section 5 with at least one concrete application of the tilt-invariance result.

* The equivalence with the Orlicz norm is briefly mentioned, but not elaborated upon. Since Orlicz norms are often used in concentration inequalities and learning theory it might help to more clearly explain this connection, especially for readers less familiar with the $q$-subweibull literature.

---

### Decision · Action_Editor_XtNa · 2025-09-03

**Recommendation:** Accept with minor revision

**Additional Comments:**

As requested by Reviewer u7qH, since this is a rather short paper, it would be great for the author to add some explicit motivation and justification (even if it's just a paragraph or a short section) on why the results here (especially the ones on tilting) are of interest to a part of the TMLR community.

Another typo I saw: "perversed" -> "preserved" in the second to last sentence in Section 6.

**Audience:**

Yes

**Audience Explanation:**

Results on tails and concentration of (interesting classes of) random variables are some of the more basic bread-and-butter tools in learning theory. Sub-Weibull distributions are an important class, covering many common distributions (e.g. Poissons and standard sub-Gaussians). This paper provides a good reference on this important class.

**Claims And Evidence:**

Yes

**Claims Explanation:**

The mathematical claims are clearly stated, and accompanied by proofs. Reviewers have no concern about the validity of the results.